# PD-1/PD-L1 Immuno-Mediated Therapy in NAFLD: Advantages and Obstacles in the Treatment of Advanced Disease

**DOI:** 10.3390/ijms23052707

**Published:** 2022-02-28

**Authors:** Rosa Lombardi, Roberto Piciotti, Paola Dongiovanni, Marica Meroni, Silvia Fargion, Anna Ludovica Fracanzani

**Affiliations:** 1General Medicine and Metabolic Diseases, Fondazione IRCCS Ca’ Granda Ospedale Maggiore Policlinico, Pad. Granelli, Via F Sforza 35, 20122 Milan, Italy; rosa.lombardi@unimi.it (R.L.); roberto.piciotti@unimi.it (R.P.); paola.dongiovanni@policlinico.mi.it (P.D.); marica.meroni@policlinico.mi.it (M.M.); silvia.fargion@unimi.it (S.F.); 2Department of Pathophysiology and Transplantation, University of Milan, 20122 Milan, Italy

**Keywords:** NAFLD, NASH, liver immune microenvironment, PD-1, immune therapy

## Abstract

Non-alcoholic fatty liver disease (NAFLD) is characterized by an enhanced activation of the immune system, which predispose the evolution to nonalcoholic steatohepatitis (NASH) and hepatocellular carcinoma (HCC). Resident macrophages and leukocytes exert a key role in the pathogenesis of NAFLD. In particular, CD4+ effector T cells are activated during the early stages of liver inflammation and are followed by the increase of natural killer T cells and of CD8+ T cytotoxic lymphocytes which contribute to auto-aggressive tissue damage. To counteract T cells activation, programmed cell death 1 (PD-1) and its ligand PDL-1 are exposed respectively on lymphocytes and liver cells’ surface and can be targeted for therapy by using specific monoclonal antibodies, such as of Nivolumab, Pembrolizumab, and Atezolizumab. Despite the combination of Atezolizumab and Bevacizumab has been approved for the treatment of advanced HCC, PD-1/PD-L1 blockage treatment has not been approved for NAFLD and adjuvant immunotherapy does not seem to improve survival of patients with early-stage HCC. In this regard, different ongoing phase III trials are testing the efficacy of anti-PD-1/PD-L1 antibodies in HCC patients as first line therapy and in combination with other treatments. However, in the context of NAFLD, immune checkpoints inhibitors may not improve HCC prognosis, even worse leading to an increase of CD8+PD-1+ T cells and effector cytokines which aggravate liver damage. Here, we will describe the main pathogenetic mechanisms which characterize the immune system involvement in NAFLD discussing advantages and obstacles of anti PD-1/PDL-1 immunotherapy.

## 1. Introduction

Non-alcoholic fatty liver disease (NAFLD) is the commonest chronic liver disorder in Western countries and is defined by fat accumulation in more than 5% of hepatocytes in the absence of other causes of liver diseases [1]. NAFLD is usually a mild disease characterized by simple steatosis; however, it may progress to more severe forms where inflammation (non-alcoholic steatohepatitis NASH) and fibrosis occur, possibly evolving to cirrhosis and hepatocellular carcinoma (HCC) [2]. In particular, HCC is currently the fourth leading cause of cancer mortality [3] and it is estimated that NAFLD related cirrhosis and HCC will become the main indication for liver transplantation in the next years [4]. 

The pathogenesis of NAFLD is very complex and involves a “multiple hits” process where different insults act together to foster fat accumulation, inflammation and progressive liver damage [5]. Among all concurring factors, insulin resistance (IR), oxidative stress, hormones and cytokines released by adipose and liver tissues, and alterations in gut microbiota are key determinants. In addition, both genetic and epigenetic factors predispose to NAFLD and its advanced forms, including HCC [6,7]. 

Even though the pathogenesis and progression of NAFLD are multifactorial, inflammation is considered a key player in the evolution to advanced disease and HCC [8]. Indeed, the liver hosts several resident immune system cells, in particular resident macrophages called Kuppfer cells (KCs), natural killer cells (NK cells), natural killer T cells (NKT cells), and lymphocytes. In the presence of obesity, IR, and lipotoxicity, the immune system is activated in the liver, with a corresponding grade of inflammation triggered by the “wound healing response”. When inflammatory signals persist, monocytes and neutrophils are recruited in the liver, as well as other lymphocytes with an increase in CD8+ occurring and consequent further rise of the inflammatory response, activation of fibrogenesis and carcinogenesis cascade [9]. 

Different regulatory pathways counterbalance immune system activation. For example, the programmed cell death protein 1 (PD-1), a membrane protein expressed on all T cells, acts together with T-cell immunoglobulin mucin-3 (Tim-3) inhibiting their activity. Cytotoxic T-lymphocyte-associated protein 4 (CTLA-4) competes with CD28 for binding of B7 molecules and results in a net negative signal for the proliferation and the survival of the T cell [10,11]. In this regard, if on the one hand a hyperactivated immune system contributes to progression from steatosis to NASH and fibrosis, on the other hand when downregulated, it favors hepatic carcinogenesis due to insufficient tumor surveillance.

Several studies have shown that the gut microbiota plays a key role in the pathogenesis of NAFLD [12,13]. Interestingly, dysfunctions in the gut are associated with NAFLD and NASH and lead to the release of pathogen-associated molecular patterns (PAMPs) and bacterial products from enterohepatic circulation and contributes to liver inflammation [14]. 

Thus, although targeting the immune system is emerging as a promising strategy for both NAFLD and HCC treatment, several questions are still unsolved. In addition, contrasting data are accumulating in literature concerning either in vitro or in vivo models as well as in patients. 

Hence, this review aims to describe the main pathogenetic mechanisms by which the immune system fosters the development of NAFLD and its progression to advanced forms including HCC, dealing with the immune therapeutic applications of PD-1/PD-L1 target therapy. 

## 2. The Hepatic Immunological Milieu

The liver has a pivotal role in the modulation of the immune response maintaining immunotolerance to non-pathological antigens and responding to constant inflammatory stimuli. This implies the assessment of a complex immunological network and the establishment of important interactions between liver-resident cells and peripheral leukocytes. The liver, indeed, represents a milieu of innate and adaptive immune cells and accounts for the largest population of resident macrophages (Kupffer cells (KCs)) in the body, a high density of natural killer cells (NK cells), natural killer T cells (NKT cells), and of resident lymphocytes [15]. 

Moreover, compared to the periphery, a greater ratio of CD8+ T to CD4+ T cells is also reported in the liver [16]. CD8+ T cells recognize and kill cells presenting antigens through the binding of MHC class I molecules and T-cell receptor (TCR) on cell surface. For a correct activation, CD8+ Ts need a second co-stimulatory signal by B7 expressed on antigen presenting cells (APCs) binding to CD28. CD4+ T cells recognizing the peptides presented by MHC class II molecules play an important role in protecting the liver from infections but are also involved in hepatocellular damage and autoimmunity [16]. In the liver, CD4+ T cells can acquire different cell states (i.e., T_H_1, T_H_2, and T_H_17 cells) characterized by specific cytokine profiles, anyway during inflammation they can display a mixed phenotype showing simultaneously features of different polarization status [17]. 

CD8+ T and CD4+ T activity can be inhibited by specific negative regulators. In this regard, PD-1, a membrane protein exposed by all T cells, counteracts lymphocytes activation by engaging its ligands programmed cell death 1 ligand 1 (PD-L1) [18]. PD-1 is constitutively expressed in both humans and mice T cell population and its upregulation is directly promoted by interferon (IFN)-γ. PD-1/PD-L1 T cell inhibition occurs by interfering with TCR/CD28 signaling and leads to reduced pro-inflammatory cytokine production (e.g., IL-2, IFN-γ, and tumor necrosis factor (TNF)-α). 

Important participants in liver immune response are also NKT cells able to kill target cells and to produce a wide spectrum of pro- and anti-inflammatory molecules (for example, IFN-γ, IL-4, and IL-17) [19]. Additionally, liver basal immune activity is also exerted by liver sinusoidal endothelial cells (LSECs), hepatic stellate cells (HSCs) and hepatocytes which act as APCs triggering cytokine production [20]. Considering this, deregulation of liver’s tightly controlled immunological network may be a hallmark of tissue damage and may contribute to liver disease. 

New evidence claims that immune cells, both innate and adaptive, are skewed toward a proinflammatory phenotype in the liver with NAFLD and NASH [21]. Indeed, activation of immune response and the release of proinflammatory cytokines promote damaging of hepatocyte and activation of HSCs. Thus, the early stages of liver inflammation in NAFLD and NASH are dominated by CD4+ effector T cells and followed by a cytotoxic CD8+ T cell response [19,22,23]. Therefore, adaptive immune cells are deemed to have a pivotal role in NASH progression and gain particular interest as novel target for liver disease treatment. 

Immune response is also mediated by the activation of innate immune receptors in hepatocytes which trigger a proinflammatory environment found to be involved in NAFLD pathogenesis and progression [24]. Therefore, toll-like receptors (TLRs)-2, TLR4, TLR5, and TLR9, which are involved in the liver inflammation, were found activated in NAFLD and linked to the increase of some inflammation-associated nuclear transcription factors such as IRF, NF-κB and peroxisome proliferator-activated receptors (PPAR-α). Intracellular inflammasome activation in hepatocytes and cell death are also two key triggers of liver inflammation during NAFLD and NASH progression. Together they foster the production of the pro-inflammatory cytokines IL-1β and IL-18 and further enhance the downstream activation of intracellular adaptors and kinases including TNF-receptor-associated factors, TGF-β-activated kinase 1/JNK, and IκB kinases (IKKs) [9]. Furthermore, the oxidative stress in experimental obesity models have been found associated to T cell infiltration, fibrosis and HCC and is strongly promoted by STAT-1 and STAT-3 signaling in mice. 

In this regard, several papers have recently investigated, in different NAFLD/NASH preclinical models, how dysregulation of immune cells promoted by pathological metabolic changes is directly involved in tissue-damage process and in the progression to HCC. A schematic representation of T cells immune regulation upon liver disease is shown in Figure 1.

## 3. Dysregulation of the Immune System in Experimental NAFLD 

### 3.1. Fat Accumulation Induces Liver CD4+T Cell Depletion in In Vitro and In Vivo Models 

Ma et al. have recently observed the lipid-induced loss of CD4+ T lymphocytes in NAFLD patients and preclinical models supporting the tight correlation between hepatic disease and the immune micro-environment [25]. Thus, mice trans-genetically induced through liver-specific MYC oncogene (MYC-ON) and fed a methionine–choline-deficient diet (MCD) developed NAFLD and were used to assess lipid metabolism dysregulation. 

Remarkably, fewer CD4+ T lymphocytes, but not CD8+ T cells, were found in MYC-ON MCD mice compared to heathy controls. As cause, CD4+ T lymphocyte death was found to be induced by linoleic acid (C18:2) accumulation within the cells. Dose–response and time-course analysis revealed that CD4+ T lymphocytes were more susceptible to linoleic acid -induced cell death than CD8+ T lymphocytes and culturing of CD4+ T lymphocytes with free fatty acids (FFA)-depleted conditioned medium no longer caused their death in MYC-ON MCD mice. 

Dysregulation in mitochondrial-derived ROS production and in beta-oxidation were also identified as the critical mediator for CD4+ T lymphocyte death. Carnitine palmitoyltransferase 1 (CPT1), the rate-limiting enzyme in long-chain fatty acid oxidation (FAO), was found increased within CD4+ T mitochondria, in parallel with a decreased expression of several genes coding for components of the electron transport complex (ECT). As consequence, mitochondrial membrane potential, maintained by proper ETC activity, was significantly decreased by linoleic acid in CD4+ but not CD8+ T lymphocyte prompting ROS generation and resulting in CD4+ T cell death. Furthermore, after C18:2 treatment, the increase of both mitochondrial superoxide and caspase 3/7 activity confirmed that CD4+ T lymphocytes died through mitochondrial ROS-induced apoptosis. As proof, CPT1 knockdown blocked linoleic acid-induced mitochondrial ROS production in CD4+ T cells and the blockage of ROS production by using N-acetylcysteine (NAC) abrogated cell death in vitro in CD4+ T lymphocytes when incubated with hepatocytes from MCD-ON MYC mice and reversed in vivo the loss of hepatic CD4+ T lymphocytes in MCD-diet-fed mice.

### 3.2. CD8+ T Cells and NKT Cells Mediate Liver Damage in In Vitro and Murine Models: An Insight into the PD-1/PD-L1 Signaling Pathway

In in vitro and preclinical models which recapitulate the key features of NASH, it has been assessed the hepatic accumulation of CD8+ T cells and their functions, thus unveiling their pivotal role in liver disease progression. In a recent study, Dudek et al. demonstrated that in NASH mice CD8+ T liver cells are involved in auto-aggressive tissue damage through defined, non-redundant sequential activation steps [26]. NASH was induced in animals through the administration of a choline-deficient, high-fat diet (CD-HFD) for 9–12 months. Subsequently, chemokine receptor CXCR6, a prominent marker for liver residency and for tissue-resident and effector memory, was found overexpressed in CD8+ T cells and co-expressed with markers associated with CD8+ T exhaustion (such as PD-1) and indicative for their effector function such as Granzyme B (GzmB), TNF and IFN-γ [27,28]. 

Genome-wide transcriptome analysis and gene-set enrichment analysis (GSEA) on NASH liver CXCR6^+^ CD8+ T cells have furtherly confirmed these findings in mice. Thus, genes associated with T cell effector function (as *Bhlhe40* which encodes for granzymes) and activation, exhaustion (*Pdcd1* and *Tox*) and tissue residency (*S1pr1*, *Klf2,* and *Rgs1)* were found overexpressed in this setting. FOXO1 was demonstrated through loss and gain-of-function experiments to be an effective negative regulator of CXCR6 expression in CD8+ T. Namely, the inhibition of FOXO1 caused the upregulation of CXCR6, whereas the overexpression of FOXO1 reduced the expression of CXCR6 in CXCR6^−^ CD8+ T. In turn, *Foxo1* expression is regulated by IL-15 signaling which mediates FOXO1 phosphorylation allowing the nuclear export and proteasomal degradation of cytosolic FOXO1 [29]. Notably, both serum and mRNA levels of IL-15 were higher in patients and in mice with NASH suggesting the pivotal role of IL-5 in the induction of CXCR6+ CD8+ T cells. As proof, treatment with IL-15 downregulated FOXO1 in mice CD8+ T cells thus, upregulating CXCR6 and other CD8+ T inhibitors such as PD-1, GzmB and CD69 which are involved in the immune response. Moreover, serum IL-15 levels were found to be correlated with higher frequency of hepatic CXCR6^+^PD-1^high^ CD8+ T cells and with low FOXO1 expression. In this setting, low FOXO1 activity rendered T cells responsive to environmental metabolic cues such as acetate and capable of killing hepatocytes triggering auto aggression and boosting chronic liver damage. 

Accordingly, Wolf et al. claimed that CD8+ and NKT cells cooperatively induce liver damage in mice fed CD-HFD enhancing steatosis, NASH development, and transition to HCC. In particular, NKT cells mediate the lipid uptake via lymphotoxin β receptor (LTβR) activation in hepatocytes inducing fat accumulation and leading to the onset of macrovesicular steatosis [23]. 

## 4. Dysregulation of the Immune System in Human NAFLD

### 4.1. Hepatic CD4+T Cell Depletion Characterizes Human NAFLD 

Supported by the evidence in NAFLD experimental models, studies on the modulation of the inflammatory cascade are accumulating also in humans. In particular, they focused on the immune pathogenetic mechanisms of NAFLD and its progression to advanced forms, especially HCC. 

As described above, in murine models with NAFLD a dysregulation of lipid metabolism due to the accumulation of linoleic acid induced the CD4+ T lymphocyte death, with consequent alteration in the immune system balance within the liver and promotion of hepatic damage [25]. 

The same Authors confirmed these data also in lymphocytes of peripheral blood of patients with NAFLD, as the exposure to linoleic acid led to a selective death of CD4+ but not CD8+ lymphocytes mediated by increased oxidative stress. In addition, analyzing the concentration of CD4+ cells in biopsies specimens of patients with NASH, alcoholic or viral disease, a fewer count was found in NASH subjects, confirming the depletion of CD4+ caused by hepatic fat. 

As in murine models, death of CD4+ seems to be caused by oxidative stress. In fact, Feldstein et al. showed in plasma of 122 patients with liver histology that products of free radical-mediated oxidation products of linoleic acid measured by mass spectroscopy were significantly elevated in patients with NASH compared to those with simple steatosis. In addition, a positive correlation between the amount of oxidation products and severity of liver histologic features (mainly inflammation and fibrosis) was demonstrated [30]. 

### 4.2. CD8+ T Cells and NKT Cells Mediate Liver Damage in Human NAFLD: An Insight into the PD-1/PD-L1 Complex

As previously discussed, Wolf et al. demonstrated that CD8+ and NKT cells cooperatively induce liver damage in fed CD-HFD mice enhancing steatosis, NASH development, and transition to HCC. Again, analyzing liver histological specimens of patients with NASH, alcoholic and viral diseases and healthy controls, the authors showed a higher number of CD8+ and NKT in the samples of NASH subjects, as well as NASH induced HCC compared to healthy subjects [23]. Therefore, similarly to what observed in murine models, it could be speculated that also in patients with NASH exists an interaction between CD8+ T cells, NKT cells, their secreted cytokines and hepatocytes, thus suggesting that targeting specific immune signaling pathways could potentially diminish the risk to develop liver damage and progression to HCC. 

Given the strict association between CD8+ and PD-1/PD-L1, recent studies focused on this signaling pathway, although the results are conflicting and not conclusive. Polymorphisms in the PD-1 gene have been associated with an increased risk of various types of cancers and some of them alter protein expression and function [31]. The PD-1 rs10204525 C > T and rs7421861 A > G variants boost PD-1 expression and have been associated with increased risk of esophageal cancer in Asian individuals [32]. Furthermore, Kaplan-Meier survival curves showed that higher PD-1 gene expression contributed to worse survival of esophageal cancer patients [32].

In a cohort of 594 patients with NAFLD and 391 with NAFLD-HCC who were enrolled from three European sites (UK, Milan, and Berna), the PD-1 rs7421861 was independently associated with NAFLD-HCC only in the UK cohort thus suggesting that the impact of genetic variants which modify the HCC milieu may differ according to ethnicity although pathways may be shared [33,34].

In a cohort of 134 Egyptian patients with biopsy proven NASH and NASH related HCC, the rs2282055 G > T genetic variant in the PD-L1 gene was associated with cancer [35]. In another study involving 167 HCC patients, PD-L1 hepatic levels were increased, paralleling CD8+ expression, and IFN-γ was found to be positively correlated with CD8+ and PD-L1 gene expression. These findings suggest that PD-L1 is not constitutively expressed by tumor cells but it may represent an adaptive mechanism of them to escape endogenous antitumor activity. However, there was no association between PD-L1 levels and HCC severity. Conversely, this study highlighted a better survival in patients with higher intra-tumoral expression of PD-L1 and CD8+, confirming the role of the cytotoxic T cell in the eradication of HCC [36]. In line with this report, Sideras et al. confirmed that low expression of PD-L1 and low CD8+ predict extremely poor HCC-specific survival in a cohort of 154 patients with resected liver tumors [37]. 

These results contrast with other evidences in smaller cohorts, where PD-L1 overexpression was associated with a poor prognosis in patients with HCC [38,39,40].

### 4.3. PAMPs Foster Liver Inflammation by Activating Immune Response in NAFLD Patients 

PAMPs, activate the pattern recognition receptors (PPRs) in the liver further contributing to hepatic inflammation and liver disease progression. Patients with NAFLD with pronounced intestinal inflammation show decreased numbers of CD4+ and CD8+ T lymphocytes in the intestinal mucosa, associated with increased cytokine secretion and disruption of tight junctions [14]. Lipopolysaccharide (LPS), a PAMP which activates TLR4 signaling, induces liver inflammation triggering the production of tumor necrosis factors by liver macrophages and participates in the development of hepatic fibrosis [41]. Moreover, it has been previously demonstrated that the response of colonic myofibroblasts (CMFs) to the bacterial PAMPs may favor a suppressive effect on activated CD4+ T cells in the colonic mucosa via upregulation of negative co-stimulator PD-L1 mainly through TLR2, 4, and/or TLR5. Thus, proved that PD-1/PD-L1 may be expressed on DC cells, which are highly proficient APCs in the liver and steadily exposed to intestinal PAMPs in order to decelerate immune responses and to prevent inflammation [42]. Therefore, although scientific papers about the correlation between PAMPs and PD1/PDL1 axis in the context of NAFLD are still not present in literature, it could be speculated that the two signaling pathways might interplay in liver diseases.

## 5. Immune-Related Therapy in NAFLD 

Reinforced by studies supporting a central role for the immune system in the regulation of NAFLD pathogenesis and its progression towards advanced forms, as well as carcinogenesis, a modulation of this system as therapeutic attempt was expected. Different possible targets have been selected, namely, chemokine receptors, nuclear receptors, molecules expressed on immune system cells surface, tyrosin-kinases and anti-angiogenetic agents. Among all, the PD-1/PD-L1 complex has received great attention. However, while trials with drugs targeting this complex in HCC are accumulating, on the contrary data are available on the application of PD-1/PD-L1 blockage to prevent NAFLD progression to NASH or fibrosis. 

### 5.1. Immunotherapy for NAFLD and Hepatic Fibrosis

Nowadays no pharmacological therapy has been approved for NAFLD, especially in the advanced forms of NASH and fibrosis [43]. However, since the immune response seems to play a key role in NASH and in its progression to HCC, several trials have explored the possibility to address it to treat patients. 

An experimental drug, Cenicriviroc (CVC), showed the ability of targeting proinflammatory monocytes via the dual CCR2/CCR5 chemokine receptors antagonist firstly in murine models and then in a phase IIb clinical trial (CENTAUR trial) which described an improvement in fibrosis in biopsied patients with NAFLD [44]. However, these encouraging data were not confirmed in the phase III trial (AURORA) which was prematurely interrupted because the primary endpoint (i.e., improvement of fibrosis without worsening of NASH) was not achieved evaluating interim analysis. 

In another phase II trial (TANDEM), CVC was combined with Tropifexor, a farnesoid X receptor agonist (FXR), for the improvement of hepatic fibrosis in the same category of patients, although the analysis of results is still ongoing [45]. FXR is a member of the nuclear receptor superfamily and is mainly expressed in the liver, intestine, kidney, adipose tissue, and immune cells and is involved in bile acid synthesis and transport, glucose and lipid metabolism and exerts an anti-inflammatory and anti-fibrotic action [46]. 

Other trials targeted macrophages to exert an anti-inflammatory switch mediated by a peroxisome proliferator-activated receptors (PPARs) agonists [47]. PPARs are nuclear receptors which activation regulates lipid and glucose metabolism as well inflammation though macrophage modulation and gene expression in different tissues, included the liver. Furthermore, they are involved in maintaining quiescent the hepatic stellate cells, thus preventing fibrogenesis. Different molecules have been tested (i.e., Fenofibrate, Thiazolinedions, Seroglitazar, Seldalepar); however, none of them resulted definitely effective in improving liver histology. In particular, despite enthusiastic results in the phase II study, Elafibranor, a dual PPAR alfa/delta agonist, failed to confirm its beneficial effect in the phase III trial [47]. 

In another phase IIa clinical trial, patients with NASH and type 2 diabetes mellitus were treated with a murine monoclonal antibody (OKT3) which targets T cell receptor-associated molecule CD3, with consequent improvement in transaminases and IR [48]. However, murine anti-CD3 Ab stimulates an extensive release of cytokines within the initial hours after the first administration with high toxicity, thus limiting its clinical use. A new fully human anti-CD3 antibody has been developed, called foralumab and already tested in Chron Disease and in renal allograft reject. Currently, a phase II clinical trial is ongoing (NCT03291249) to determine safety and efficacy of foralumab in patients with NASH and T2DM [49]. However, despite several attempts in the modulation of immune system for controlling the pathogenesis and progression of NAFLD a conclusive evidence is still lacking.

### 5.2. Immunotherapy for Hepatocellular Carcinoma

Since HCC is emerging as one of the leading causes for cancer-related mortality, great efforts have been performed to identify an efficient curative strategy. Currently, the only proposed strategies for HCC are loco-regional treatments (i.e., hepatic resection, ablation, transarterial chemoembolisation), inhibitors of tyrosin kinase/anti-angiogenetic agents, as Sorafenib, Regorafenib and ramucirumab, and ultimately liver transplantation [50]. Very recently also Lenvatinib, an oral multikinase inhibitor, was confirmed as non-inferior to Sorafenib in untreated advanced hepatocellular carcinoma [51,52].

However, if on the one hand data on the efficacy of immunotherapy in patients with NASH and severe fibrosis are not conclusive yet, the results obtained in patients with HCC, developed on liver disease of different etiology, seem more promising. Indeed, given the established role of the immune system in carcinogenesis, in recent years immunotherapy, and in particular the immune checkpoint blockade, has been introduced in the treatment of HCC and may improve the prognosis of liver disease [53]. In this regard, checkpoint inhibitors such as anti-PD-1/PDL-1 antibodies have been used to restore immune control of tumors by disrupting co-inhibitory receptors and enhancing anti-tumor T cell responses. 

Despite the pathophysiological bases of this mechanism are clear and support the use of the PD-1/PD-L1 blockade, data in clinical practice are contrasting and to date the relationship between PD-L1 expression and host-tumor immunity in HCC is not well defined. 

However, even if immunotherapy has been tested in this setting either as first line therapy or as adjuvant therapy after curative interventions, it may also contribute to “uncheck” the immune system to control HCC, thus worsening the underlying NASH.

As first line therapy, different monoclonal antibodies anti PD-L1 have been tested as camrelizumab, cemiplimab, and avelumab [54,55,56]. All of them have been experimented in phase I-II studies in patients previously exposed to systemic therapies and in small cohorts. 

More promising results were expected for nivolumab, pembrolizumab and atezolizumab. These PD-1/PD-L1 inhibitors have been combined with anti-antiangiogenic agents, such as Lenvatinib, Apatinib, and bevacizumab in phase I and II studies. Indeed, considering the extensive vascularization found in tumor as HCC, this association finds its application. The only phase III study in this context was the IMBRAVE-150, aimed to test the combination of atezolizumab with bevacizumab. Enthusiastic results at the phase I of the study supported this therapy regimen, leading to improvement in the median OS of 5.8 months and the PFS of 2.6 months when compared to Sorafenib, with similar rates of adverse events [57]. As for the phase II, the most promising evidence was provided in the RESCUE trial where a median overall survival of 74.7 months was reported after testing the association between camrelizumab + apatinib in a cohort of 190 patients [58]. Therefore, this therapy has been approved for patients with unresectable HCC and without prior systemic therapy [59], but a subgroup analysis did not show a significantly improvement in survival of patients with HCC arise on non-viral (NASH) vs. viral liver disease. These results suggested a possible impact of the etiology of liver disease and consequently of the hepatic microenvironment on the therapy outcome, as confirmed by further evidence as discussed below. 

Conversely, either the KEYNOTE-240 trial which evaluated the overall survival in patients who received pembrolizumab and previously treated with Sorafenib and the CheckMate-459 one which compared the survival of patients treated with nivolumab vs. Sorfenib, failed to give positive results [60,61], even though firstly obtained in phase I/II studies as depicted in Table 1. It is important to underlie that these trials showed a response only in a certain proportion of patients, mainly those with a viral etiology of liver disease and possibly in those in whom increased expression of PD-L1 were associated with an abundant CD8+ infiltration with consequent neoplasm control [62].

Recently, the role of nivolumab in HCC was also explored in combination with ipilimumab, a target specific monoclonal antibody against CTLA-4. The combination of the two immune checkpoint inhibitors was approved by US FDA in patients who were in-tolerant to sorafenib or with advanced disease after the promising results achieved in the cohort 4 of the CheckMate-040 trial. Therefore, this regimen achieved an overall response rate (ORR) of 32% (95% CI, 20–47) and an increase in terms of survival, with a median OS of 22.8 months (95% CI, 9.4-NR) versus 12.5 months (95% CI, 7.6–16.4) compared with the other arms of the study. To date, this therapy setting is furtherly under investigation in the phase III CheckMate 9DW trial (NCT04039607) and is tested in first line against sorafenib [63]. Monotherapy with anti-CTLA-4 factors has still not been tested in HCC based on the low response rates found in the latest clinical trials conducted in other cancer types. Conversely, combination of CTLA-4 and PD-1 blockade are supposed to act synergistically and to be involved in the inhibition of effector T-cell and NK cell activation in peripheral tissues and in induction of Treg cell differentiation. Therefore, when combined they results in increased response rates and higher survival rates [64]. 

To date, no adjuvant treatment has been proven to improve survival of patients with an early-stage HCC following loco-regional treatments or resection. Currently different ongoing phase II and III clinical trials, resumed in Table 1, are testing the efficacy of nivolumab (CheckMate 9DX, NCT03383458), pembrolizumab (KEYNOTE-937, NCT03867084), atezolizumab plus bevacizumab (IMbrave-050, NCT04102098), although results are still not available [65].

**Table 1 ijms-23-02707-t001:** List of the main phase II and III clinical trials for administration of immunotherapy in the context if HCC.

Clinical Trial	Phases Study Name	N Patients	Interventions	Results	Ref.
First line therapy in sorafenib non-experienced patients					
(NCT03434379)	III IMBRAVE 150	501	atezolizumab + bevacizumab vs. Sorafenib	Median PFS (HR, 95% CI): 6.8 vs. 4.3 months (0.65, 0.53–0.81; *p* = 0.0001) Median OS (HR, 95% CI): 19.2 vs. 13.4 months (0.66, 0.52–0.85; *p* = 0.0009)	[66]
(NCT02576509)	III CheckMate 459	743	nivolumab vs. sorafenib	Median PFS (HR, 95% CI): 3.7 vs. 3.8 months (0.93, 0.79–1.10; NS) Median OS (HR, 95% CI): 16.4 vs. 14.8 months (0.85, 0.72–1.00; *p* = 0.052)	[67]
(NCT02715531)	I b GO30140	59	atezolizumab	Median PFS: 3.4 months	[57]
				Median OS: N/A	
		164	atezolizumab + bevacizumab	Median PFS: 7.3 months	
				Median OS: 17.1 months	
(NCT03006926)	I b KEYNOTE-524	104	pembrolizumab + lenvatinib	Median PFS: 9.3 months Median OS: 22 months	[68]
Second line therapy in sorafenib-experienced patients					
(NCT02702401)	III KEYNOTE-240	413	pembrolizumab vs. placebo	Median PFS (HR, 95% CI): 3.0 vs. 2.8 months (0.72, 0.57–0.90; *p* = 0.002) Median OS (HR, 95% CI): 13.8 vs. 10.6 months (0.78, 0.61–1.00; *p* = 0.024)	[60]
(NCT02702414)	II KEYNOTE-224	224	pembrolizumab	Median PFS: 4.9 months Median OS: 12.9 months	[69]
(NCT01658878)	I/II CheckMate 040	145	nivolumab	Median PFS: 4 months	[62]
				Median OS: 15.6 months	
		50	nivolumab + ipilimumab	Median PFS: N/A	[70]
				Median OS: 22.8 months	
				Median PFS:2.1 months (2–3.4)	[54]
(NCT02989922)	II	217	camrelizumab	Median OS: 13.8 months (11.5–16.6)	
				Median PFS: 3.7 (2.3–9.1)	[55]
(NCT02383212)	I	26	cemiplimab	Median OS: N/A	
				Median PFS: 4.4 months	[56]
(NCT02519348)	II	40	avelumab	OS: 14.2 months	

OS: Overall Survival; PFS: Progression-free survival.

### 5.3. HCC in the Context of NAFLD May Not Benefit of Checkpoint Inhibition

As discussed above, emerging data identified in NASH an unfavorable prognostic factor towards response to this type of immuno-therapy compared to liver diseases which precede cancer onset deriving from different etiologies. 

This hypothesis is supported by evidence driven by experimental models where treatment with nivolumab and pembrolizumab gave poor results in mice with NASH characterized by PD-1 overexpression. Pfister et al. confirmed this data by performing single cell RNA-seq analysis in leukocytes population of diet-induced NASH mice [71] where they observed an increased number of CD8+PD-1+ T cells expressing markers of exhaustion, suggesting their involvement in liver damage. In concordance, mice depleted of CD8+ T cells had a reduced liver damage and lower incidence of HCC. Furthermore, an increase of PD-L1 expression in hepatocytes and non-parenchymal cells was also found and directly correlated with the severity of liver disease. In line with these data, the administration of anti-PD-1 immunotherapy aggravated liver damage and increased the number of hepatic CD8+PD-1+ T cells. In keeping with these findings, a higher number of activated hepatic CD8+ T cells which express effector cytokines (IFN-γ, TNF-a) together with both progressive liver damage and HCC incidence were observed in *Pdcd1−/−* mice suggesting the unfavorable effects of anti-PD-1 treatment in NASH which support the tissue-damage role exerted from CD8+PD-1+ T lymphocytes.

Despite the negative impact of NAFLD/NASH on the response to immunotherapy has been confirmed also in pharmacological trials on patients with HCC, a recent meta-analysis considering 3 phase III trials comprising a total of 1656 patients with unresectable HCC found that immunotherapy improved overall survival only in patients with viral HCC compared to non-viral ones [71]. This evidence was sustained by another meta-analysis involving eight studies comprising more than 3700 patients [72]. Even worse, the checkpoint inhibition of PD-1/PD-L1 was linked to a decreased survival in patients NAFLD related HCC [71]. Conversely, no differences related to etiology were observed in patients treated with tyrosine kinase inhibitors /anti–vascular endothelial growth factor. 

## 6. Concluding Remarks

In conclusion, both data on experimental models and patients confirmed the crucial role of the immune system and inflammatory pathways in the pathogenesis and progression of NAFLD to advanced disease and HCC. In particular, in NAFLD/NASH models a harmful role for CD8+ has been established, so that attention in the scenario of NAFLD therapy has been given to the PD-1/PD-L1 signaling pathway, which negative regulate lymphocyte cytotoxic action. However, if on the one hand turning off the immune system activation could block damage progression in NAFLD, on the other hand, it may weaken immune system tumor surveillance, thus promoting carcinogenesis. Despite the interesting topic, no data are available on the application of PD/PD-L1 blockage to prevent NAFLD progression to NASH or fibrosis, whereas immunotherapy and in particular the immune checkpoint blockade of PD-1/PD-L1, has been experimented in the treatment of HCC. Among all possible drugs targeting this complex, the combination of atezolizumab and bevacizumab and the combination of nivolumab with ipilimumab have been approved for the treatment of advanced HCC. Despite promising results in the therapy of HCC, it should be taken in mind that patients with NAFLD related HCC may be not the ideal candidates for this innovative therapy given opposite results in NASH progression and HCC control. 

## Figures and Tables

**Figure 1 ijms-23-02707-f001:**
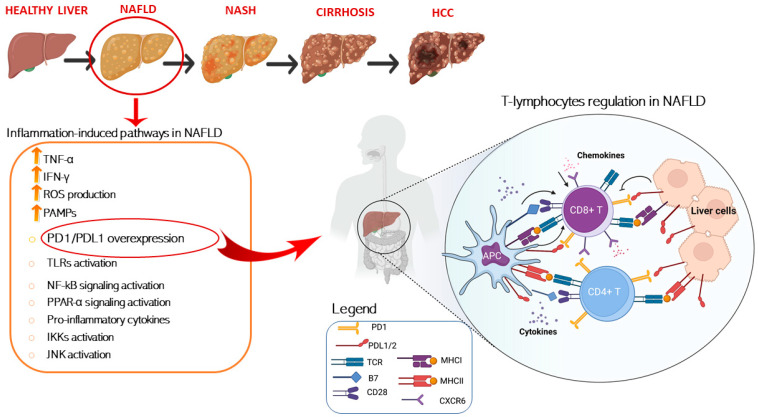
Schematic representation of T cells immune regulation and inflammation-induced pathways upon liver disease. Liver inflammation is a key factor in NAFLD pathogenesis and is characterized by increased levels of several inflammation associated molecules, such as TNF-α and IFN-γ and inflammation-induced molecular pathways activation. T cell activation requires two signals, the first signal is specific and implies T cell receptor recognition and binding to MHC/antigen presented by an antigen-presenting cell or by tissue specific immunized cells. The second signal is nonspecific and involves the B7 ligand, exposed on the antigen-presenting cells to bind its receptor, CD28, on T cells. On the other hand, T cell are mainly inhibited trough the binding of the exhaustion marker PD-1 which is expressed on cell surface binding to its ligand PD-L1/2 present on APC and over-exposed on liver cell in HCC setting. Conversely, T cell activation is damped by increased levels of immunosuppressive cytokines such as IL-4, IL-8, and IL-10. Instead, IL-5 release induces chemokine receptor CXCR6 upregulation by increasing CD8+ T cells susceptibility to metabolic stimuli and triggering CD8+ auto-aggression of liver tissue. Antigen presenting cells (APCs); programmed cell death protein 1 (PD-1); programmed cell death 1 ligand 1/2 (PD-L1/2); T-cell receptor (TCR); major histocompatibility complexes I/II (MHC I/II); pathogen-associated molecular patterns (PAMPs); Toll-like receptors (TLR); nuclear factor kappa (NF-kB); peroxisome proliferator-activated receptors (PPAR-α); IκB kinases (IKK); c-Jun N-terminal kinases (JNK), tumor necrosis factor alpha (TNF-α), interferon gamma (IFN-γ).

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
