# Peer review of "PD-1/PD-L1 Immuno-Mediated Therapy in NAFLD: Advantages and Obstacles in the Treatment of Advanced Disease"

_ijms, 2022, doi:10.3390/ijms23052707_

Round 1

Reviewer 1 Report

This article discuss the main pathogenic mechanisms, which characterize the immune system involvement in NAFLD and the advantages and obstacles of anti PD-1/PDL- 1 immunotherapy. It seems that targeting the immune system is emerging as a promising strategy for both NAFLD and HCC treatment.

Hepatic CD4+T cell depletion characterizes human NAFLD, which will generate a dysregulation of the immune system in human NAFLD.

The article is well written and has possible clinical implications in the future. Therefore, it deserve to be published.

I would propose to summarize into a scheme the pathways of NAFLD evolution to HCC (including the PD-1/PD-L1 signaling pathways) and the experimental and clinical therapeutic targets (similar to figure 1).

Author Response

Included as a rebuttal letter file the Riviewer 1 can find the answers to the required questions

Reviewer 2 Report

This is a well written review on the PD-1/PD-L1 axis from both pathological and therapeutical viewpoints in the context of NAFLD. The below list is some concerns and requests for the improvement of this manuscript.

  • lines 109–115: Although this article focuses on the PD-1/PD-L1 axis, CTLA4 and TIM3 may be briefly mentioned elsewhere. Also, the authors may want to explain why they chose the PD-1/PD-L1 axis, but not CTLA4 or TIM3, as a main topic of this review article. Are there any differences in their pathological roles in terms of NASH and NASH-related HCC?

  • Infiltration of LPS or other pathogens produced by gut microbiota through portal vein is suggested as an additional causative factor for the dysregulated immune system in NAFLD. I would like the authors to state whether PAMPs affects the PD-1/PD-L1 axis in the liver.

  • line 300: X receptor (FXR) "agonist" (please insert "agonist")

Author Response

Included as a rebuttal letter file the Reviewer 2 can find the answers to the required questions

Reviewer 3 Report

Authors reviewed immuno-pathogenesis of NAFLD and discussed immunotherapy in NAFLD. This topic was interesting and important for clinicians. But the content was unclear and issues remained to be addressed. In figure 1, it was the data on NAFLD. Authors should summarize the figure to show the mechanism inn NAFLD. A lot of recent studies in real-world setting have focused on immunotherapy for HCC with NAFLD. Authors should discuss about them.

Author Response

Included as a rebuttal letter file the Reviewer 3 can find the answers to the required questions

Round 2

Reviewer 1 Report

Thank you for responding to my comments.

Reviewer 3 Report

Revised manuscript was well addressed to reviewers' comments and well written. It is useful for clinicians to understand the pathogenesis and future perspectives.